# Virulence, multiple drug resistance, and biofilm-formation in *Salmonella* species isolated from layer, broiler, and dual-purpose indigenous chickens

Sicelo B. Dlamini[1,2]*, Victor Mlambo[2], Caven Mguvane Mnisi[1,3], Collins Njie Ateba[4]

1 Department of Animal Science, School of Agricultural Sciences, North-West University, Mafikeng, South Africa, 2 School of Agricultural Sciences, Faculty of Agriculture and Natural Sciences, University of Mpumalanga, Nelspruit, South Africa, 3 Food Security and Safety Focus Area, Faculty of Natural and Agricultural Sciences, North-West University, Mafikeng, South Africa, 4 Department of Microbiology, Faculty of Natural and Agricultural Sciences, North-West University, Mafikeng, South Africa

* Sicelo.Dlamini@ump.ac.za

**Data Availability Statement:** All relevant data are within the manuscript and its Supporting Information files.

## Abstract

Globally, the significant risk to food safety and public health posed by antimicrobial-resistant food-borne *Salmonella* pathogens is driven by the utilization of in-feed antibiotics, with variations in usage across poultry production systems. The current study investigated the occurrence of virulence, antimicrobial resistant profiles, and biofilm-forming potentials of *Salmonella* isolates sourced from different chicken types. A total of 75 cloacal faecal samples were collected using sterile swabs from layer, broiler, and indigenous chickens across 15 poultry farms (five farms per chicken type). The samples were analysed for the presence of *Salmonella* spp. using species-specific PCR analysis. Out of the 150 presumptive isolates, a large proportion (82; 55%) were confirmed as *Salmonella* species, comprising the serovars *S.* typhimurium (49%) and *S.* enteritidis (30%) while 21% were uncategorised. Based on phenotypic antibiotic susceptibility test, the *Salmonella* isolates were most often resistant to erythromycin (62%), tetracycline (59%), and trimethoprim (32%). The dominant multiple antibiotic resistance phenotypes were SXT-W-TE (16%), E-W-TE (10%), AML-E-TE (10%), E-SXT-W-TE (13%), and AMP-AML-E-SXT-W-TE (10%). Genotypic assessment of antibiotic resistance genes revealed that isolates harboured the *ant* (52%), *tet* (A) (46%), *sui1* (13%), *sui2* (14%), and *tet* (B) (9%) determinants. Major virulence genes comprising the invasion gene *spiC*, the SPI-3 encoded protein (*misL*) that is associated with the establishment of chronic infections and host specificity as well as the SPI-4 encoded *orfL* that facilitates adhesion, autotransportation and colonisation were detected in 26%, 16%, and 14% of the isolates respectively. There was no significant difference on the proportion of *Salmonella* species and the occurrence of virulence and antimicrobial resistance determinants among *Salmonella* isolates obtained from different chicken types. In addition, neither the chicken type nor incubation temperature influenced the potential of the *Salmonella* isolates to form biofilms, although a large proportion (62%) exhibited weak to strong biofilm-forming potentials. Moderate to high proportions of antimicrobial resistant pathogenic *Salmonella* serovars were detected in the study but these did not vary with poultry production systems.

**Funding:** SB Dlamini (Grant no: 138276) This work was supported by the National Research Foundation (NRF) Grant no: 138276. The URL of the funders website is https://www.nrf.ac.za The funders did not play any role in the study design, data collection and analysis, decision to publish, or preparation of the manuscript.

## 1. Introduction

The soaring demand for poultry products has seen production methods shift from extensive to intensive techniques to maximise bird productivity [1]. However, intensive production systems are highly stressful environments that compromise bird immune function resulting in high incidences of disease and poor growth performance. To mitigate the negative impacts of stress and infectious diseases, producers have traditionally relied on the use of antibiotics for growth promotion and therapeutic processes [2]. Unfortunately, the extensive use of antimicrobial agents contributes to the emergence of antimicrobial resistance [3], which threatens poultry production, food safety, and public health [4].

Antimicrobial resistance (AMR) has been reported in *Salmonella* species, thus contributing a significant risk to food safety and public health globally [5]. The most prevalent zoonotic *Salmonella* serotypes, comprise *Salmonella enterica* serotype Enteritidis and *Salmonella enterica* serotype Typhimurium, and these are responsible for salmonellosis in humans and animals [6]. While some *Salmonella* serotypes cause self-limiting gastroenteritis in humans, resistant cells most often are associated with more complicated infections, which are challenging to treat especially in vulnerable groups such as children [7], the elderly, and immunocompromised individuals [8]. Although it is generally accepted that *Salmonella* species occur in the gastrointestinal tract of chickens [9], investigations into the influence of poultry production systems and associated husbandry practices on their prevalence are rather limited. Several studies conducted on *Salmonella* species in poultry from the study area focused on the prevalence of virulent and AMR *Salmonella* serovars [10–12], with no consideration of the influence of poultry production systems and management practices. The prevalence and distribution of AMR pathogens in chicken vary with management practices and biosecurity standards across different poultry production systems [13]. Indeed, antimicrobial resistance has been reported to occur less in free-range extensive or semi-intensive rearing systems compared to conventional production systems [14]. In addition, broilers tend to harbor more antibiotic-resistant bacteria compared to layers, which is attributed to the reduced use of antibiotics in the latter [15]. This suggests that the occurrence of AMR in indigenous chickens could be even lower than in broilers and layers because antibiotics are rarely used for native birds reared in free-range extensive production systems [16]. However, indirect transmission of antibiotic resistance among livestock animals, including indigenous chickens, especially in resource-limited settings has been reported [17, 18]. For this reason, it is necessary to monitor the distribution of AMR among different chicken types not only to achieve treatment success but also to track the emergence of AMR pathogens and possible spread to the environment and animal food products. Moreover, bacterial pathogens like *Salmonella* can develop biofilm structures (extracellular polymeric substances) and multicellular properties, enabling them to better survive chemical compounds and antimicrobial agents [19]. Recent studies have reported a high prevalence of multi-drug resistant biofilm-forming *Salmonella* serovars in poultry farms and processing facilities globally [20–24], suggesting increased risks for recurring contamination of poultry products. The potential of pathogenic bacteria including *Salmonella* species to form biofilms affects food safety even for products preserved in appropriate refrigerated storage [25]. Therefore, this study investigated the occurrence of virulence and AMR determinants, and biofilm-forming potentials of *Salmonella* serovars in intensively (layer and broiler) and semi-intensively reared (dual-purpose, indigenous) chickens in the North West province, South Africa. The study hypothesized that the prevalence of AMR and biofilm-forming potentials of *Salmonella* spp. would be higher in intensively reared compared to extensively reared birds.

## 2. Materials and methods

### 2.1 Sampling strategy

Prior to sample collection, 15 (5 broilers, 5 layers and 5 dual-purpose indigenous chickens) poultry farms were identified and selected. The owners of the farms were approached to participate in the study based on willingness. Ethical clearance for the study was obtained from the North-West University AnimCare Research Ethics Committee (approval no. NWU-00503-20-A5). Informed consent was sourced from the farmers through consent forms since no permits were required. Data on antibiotic treatment history and related husbandry practices were collected through a structured questionnaire (S1 File). The questionnaire elicited information on poultry farmers' demographic, husbandry practices, antibiotic use, and their knowledge of *Salmonella* spp. The survey instrument was face and content validated prior to administration by experts in the field of study. The reliability test of the instrument was carried out through the test-re-test reliability procedure by administering the questionnaire to two (2) poultry farmers at an interval of one week. The responses from the two administrations were then correlated and a high correlation coefficient of r = 0.80 was obtained. This confirmed the consistency and reliability of the instrument. Furthermore, a multiple contact strategy was used to eliminate sampling error and to ensure accuracy of responses gathered from the farmers. This approach eliminated the risks of receiving socially desirable responses from the farmers.

The survey results showed that broiler chicken flocks (average size: 5000 birds) were all raised intensively in housing units, most of which had foot baths for biosecurity (S2 File). Layer flocks (average size: 1000 birds) were intensively raised in battery cages. Foot baths were used at entrances of only two of the layer farms. Indigenous chicken flocks (average size: 250 birds) were raised semi-intensively with no biosecurity measures in place. Antibiotic use was highest in broilers, followed by layers while no antibiotics were used in dual-purpose indigenous chickens. Broilers and layers were fed formulated commercial diets with antibiotic growth promoters, while dual-purpose indigenous chickens mostly scavenged for their feed.

For microbiological analysis, the minimum sample size of 75 was determined to be adequate for the study using a previously reported formular [26]. To achieve this, 5 samples were collected from each farm.

### 2.2 Sample collection

A total of 75 faecal samples were collected from 15 poultry farms (5 layer, 5 broiler, and 5 indigenous dual-purpose chicken) in the Ngaka Modiri Molema District, North West province, South Africa. The faecal samples were collected directly from the cloaca of five randomly selected individual birds using sterile swabs containing multipurpose universal transport medium. The swab samples were immediately placed in the tubes containing the multipurpose universal transport medium and transported on ice to the Microbiology laboratory at the North-West University, for bacterial analysis.

### 2.3 *Salmonella* isolation

At the laboratory, swabs were immediately rinsed in 10 mL of 1% (w/v) peptone-water. After rinsing, 0.1 mL aliquots from the peptone-water were inoculated into tubes containing 10 mL of Rappaport Vassiliadis (RV) broth medium and incubated at 42˚C for 48 hrs [27]. Following enrichment, a loopful of the broth culture was streaked onto *Salmonella-Shigella* agar (SSA) plates and aerobically incubated at 37˚C for 24 hrs. Lactose non-fermenting colonies without black centres (potentially *Shigella* spp. and non-hydrogen sulphide producing *Salmonella* spp.) and lactose-fermenting colonies with large black centres (potentially hydrogen sulphide

producing *Salmonella* spp.) were randomly picked and purified on SSA. Two distinct presumptive isolates per sample were picked, thus a total of 150 presumptive *Salmonella* isolates were used for further identification tests. All pure isolates were preserved in 60% (v/v) glycerol (Merck, Johannesburg, SA) and stored at −80˚C for future use.

## 2.4 Genomic DNA extraction from presumptive isolates

Overnight cultures of presumptive isolates were prepared, and the genomic DNA (gDNA) was extracted using Zymo Research Genomic DNA$^{TM}$–Tissue MiniPrep kit (Biolab, South Africa) obtained from Inqaba Biotec, South Africa, following the manufacturer guidelines. The quality and purity of gDNA extracted from the isolates was assessed using NanoDrop Lite 1,000 spectrophotometer (model: Thermo Fisher Scientific, USA). High quality gDNA samples were stored at −80˚C for further analysis by PCR.

## 2.5 Molecular identification and confirmation of *Salmonella* isolates

Presumptive *Salmonella* isolates were subjected to *Salmonella*-specific PCR through amplification of *invA* (284 bp), *fliC* (559 bp), and *Prot6e* (185 bp) genes using a DNA thermal cycler (C1000 Touch™, BIO-RAD, South Africa) and oligonucleotides supplied by Inqaba Biotec. The primer sequence for *invA* gene was used to confirm *Salmonella* genus while *prot6e* and *fliC* genes were used to detect *S.* enteritidis and *S.* typhimurium, respectively [28]. This set of primers have been previously used to confirm the identity of the genus *Salmonella* and distinguish between *S.* typhimurium and *S.* enteriditis strains from other *Salmonella* serotypes [29]. The oligonucleotide primer sequences targeted genes, amplicon sizes and the PCR conditions (annealing temperature) are listed in Table 1. The PCR reactions constituted of 12.5 μL of 2X DreamTaq Green Master Mix, 0.5 μM of each primer, 1 μL of template DNA, and 11 μL RNase-nuclease free PCR water. A no-template DNA tube was used as a negative control while *Salmonella* Enteriditis (ATCC: 13076TM) and *Salmonella* Typhimurium (ATCC: 14028TM) reference strains obtained from Sigma Aldrich, SA were used as positive control.

## 2.6 Detection of virulence genes

Polymerase chain reaction assays were performed to amplify *spiC* (309 bp), *misL* (400 bp), and *orfL* (550 bp) virulence gene fragments. The primer sequences, targeted genes, amplicon sizes as well as the annealing temperature are listed in Table 1. All the PCR reactions were prepared in a final volume of 25 μL constituting of 12.5 μL of 2X DreamTaq Green Master Mix, 0.5 μM of each primer, 1 μL of template DNA, and RNase free water. All amplifications were performed using DNA thermal cycler (C1000 Touch™, BIO-RAD, South Africa). PCR amplicons were held at -4˚C until electrophoresis was performed.

## 2.7 Antimicrobial susceptibility test

The Kirby-Bauer disc (Mast Diagnostics, UK) diffusion technique was used to determine the antimicrobial susceptibility profile of all the *Salmonella* isolates [31]. The choice of selected antibiotics was based on the standard recommendation by CLSI, particularly antibiotics that are commonly used in the treatment of bacterial infections in both humans and animals. The list of used antibiotics comprised of gentamicin (GM10 μg), amoxicillin (A10 μg), erythromycin (E15 μg), chloramphenicol (C30 μg), tetracycline (T10 μg), trimethoprim (TM25 μg), ampicillin (AP30 μg), trimethoprim-sulfamethoxazole (TS25 μg), and kanamycin (K30 μg) [30]. The tested antibiotics belonged to 6 antimicrobial classes, which includes aminoglycosides, tetracyclines, folate pathway antagonists, phenicols, penicillins, β-lactam combination

**Table 1. Sequence of oligonucleotide primers used in PCR confirmation of *Salmonella* species and detection virulence genes [28, 30].**

| Primers | Sequences (5′ – 3′) | Target gene | Amplicon size (bp) | Annealing temperature |
|---|---|---|---|---|
| S139 | F: GTGAAATTATCGCCACGTTCGGGCAA | *InvA* | 284 | 51 |
| S141 | R: TCATCGCACCGTCAAAGGAACC | | | |
| Fli15 | F: CGGTGTTGCCCAGGTTGGTAAT | *fliC* | 559 | 55 |
| Tym | R: ACTCTTGCTGGCGGTGCGACTT | | | |
| Prot6e-5 | F: ATATGGTCGTTGCTGCTTCC | *Prot6e* | 185 | 55 |
| Prot6e-6 | R: CATTGTCCACCGTCACTTTG | | | |
| spiC | F:CCTGGATAATGACTATTGAT | *spiC* | 309 | 51 |
| spiC | R: AGTTTATGGTGATTGCGTAT | | | |
| MisL | F: GTCGGCGAATGCCGCGAATA | *misL* | 400 | 55 |
| MisL | R: GCGCTGTTAACGCTAATAGT | | | |
| orfL | F: GGAGTATCGATAAAGATGTT | *orfL* | 550 | 55 |
| orfL | R: GCGCGTAACGTCAGAATCAA | | | |

agents. A loopful of confirmed *Salmonella* isolates was inoculated into sterile nuclease free water to prepare a 0.5 MacFarland's solution of competent exponential phase growth cells. Aliquots (0.1 mL) of these cells were evenly spread-plated onto Muller Hinton agar plates [32]. Discs impregnated with CLSI recommended concentrations of the antibiotics were evenly placed on the inoculated plates and incubated aerobically at 37°C for 18 hrs. Following incubation, antibiotic growth inhibition zone diameter around the disc was measured in mm and the data was interpreted using CLSI (2023 version) guidelines. The isolates were classified as sensitive (S), intermediate resistance (I), or resistant (R) to each antibiotic following CLSI criteria [32, 34]. *Escherichia coli* ATCC 25922 was used as a reference strain because it is a recommended strain for antimicrobial susceptibility test and its quality control guidelines permit greater accuracy in interpreting AMR results [33, 34]. Percentage antibiotic resistance was calculated, and multiple antibiotic resistance (MAR) phenotypes were generated for isolates that were resistant to at least one agent in three or more antimicrobial categories [34, 35].

## 2.8 Detection of antimicrobial resistance genes

Genomic DNA of *Salmonella* extracted was used to detect antimicrobial resistance genes. All confirmed *Salmonella* isolates were screened for the presence of the *ant* (3")-la (526 bp), *tet* (A) (210 bp), *tet* (B) (659 bp), *sul1* (350 bp), and *sul2* (720 bp) antibiotic resistance determinants [30]. Primer sequences, target genes, amplicon sizes as well as PCR cycling conditions for the different genes are listed in Table 2. Polymerase chain reactions were carried out in total volumes of 25 μL each, comprising 12.5 μL of 2X DreamTaq Green Master Mix, 0.5 μM of each primer, 1 μL of template DNA and RNase free water. Amplifications were performed using DNA thermal cycler (C1000 Touch™, BIO-RAD, South Africa).

## 2.9 Phenotypic assessment of biofilm-formation

Microtiter plate assays were employed to assess the ability of *Salmonella* isolates to form biofilm at different temperatures (4°C, 25°C, and 37°C) over a 24-hour period. Triplicates of 10 μL aliquot of each overnight culture at $10^5$ CFU inoculated into 190 μL of brain-heart infusion broth per well were prepared and incubated [36]. *Pseudomonas aeruginosa* ATCC 27853 was used as a positive control because it is a strong biofilm-former. Biofilm-formation was quantified by crystal violet (CV) staining and isolates were classified into none, weak,

**Table 2. Sequences of oligonucleotide primers used for the detection of resistant antimicrobial genes in *Salmonella* isolates [30].**

| Antimicrobial agent | Sequences (5′– 3′) | Target gene | Amplicon size (bp) | Annealing temperature |
|---|---|---|---|---|
| Gentamicin | **F:** GTGGATGGCGGCCTGAAGCC | *ant (3")-la* | 526 | 60 |
| | **R:** ATTGCCCAGTCGGCAGCG | | | |
| Tetracycline | **F:** GCTACATCCTGCTTGCCTTC | *tet* (A) | 210 | 55 |
| | **R:** CATAGATCGCCGTGAAGAGG | | | |
| | **F:** TTGGTTAGGGGCAAGTTTTG | *tet* (B) | 659 | 55 |
| | **R:** GTAATGGGCCAATAACACCG | | | |
| Sulfamethoxazole | **F:** GCG CGG CGT GGG CTA CCT | *sul1* | 350 | 67 |
| | **R:** GATTTCCGCGACACCGAGACAA | | | |
| | **F:** CGG CAT CGT CAA CAT AACC | *sul2* | 720 | 52 |
| | **R:** GTG TGC GGA TGA AGT CAG | | | |

moderate, and strong biofilm-formers using automatic Enzyme-Linked Immunosorbent Assay (ELISA) microtiter plate reader (MB-580, Zhengzhou, China) [37].

## 2.10 Electrophoresis of DNA and PCR products

Genomic DNA and PCR amplicons were all separated by electrophoresis on 1.5% (w/v) agarose gel containing 0.001 µg/mL ethidium bromide using horizontal Pharmacia Biotech equipment (model Hoefer HE 99X; Amersham Pharmacia Biotech, Sweden). A 100 bp DNA molecular weight DNA marker (Thermo Fisher Scientific, South Africa) was used to confirm the sizes of the amplicons. Each electrophoresis run was conducted at 100 V for 10 min and later 80 V for 1 h using 1X TAE buffer (40 mM Tris, 1 mM EDTA and 20 mM glacial acetic acid, pH 8.0). A ChemiDoc Imaging System (Bio-Rad ChemiDoc[TM] MP Imaging System, UK) was used to capture the images using Gene Snap software, version 6.0022. Agarose gel images were analysed to determine gene sizes.

## 2.11 Statistical analysis

Optical density data were analysed using the General Linear Models procedure of Statistical Analysis System (SAS) 2010. The treatments were analyzed using a $3 \times 4$ factorial treatment arrangement in a completely randomized design according to the following model:

$$Yij = \mu + Bi + Tj + (B \times T)ij + Eijk$$

Where, $Y_{ij}$ = optical density; $\mu$ = population mean; $B_i$ = bird type; $T_j$ = incubation temperature; $(B \times T)_{ij}$ = interactive effect of bird type and incubation temperature; and $E_{ijk}$ = random error associated with observation $ijk$, assumed to be normally and independently distributed. Least squares means (LSMEANS) were compared using the probability of difference option in the LSMEANS statement of SAS.

Proportional data (arising from discrete counts) were analysed using the multinomial logistic regression procedure of SAS (2010). In the categorical variable 'bird type,' the reference category selected was broiler due to the reported extensive use of antibiotic growth promoters in this group. Consequently, there was an anticipation of a higher likelihood of detecting antimicrobial resistance in broilers. For all statistical tests, significance was declared at $P < 0.05$.

## 3. Results

### 3.1 Prevalence of *Salmonella* in different types of chickens

Thus, out of 150 isolates, 82 (55%) were confirmed to be *Salmonella* (Table 3, Fig 1). More than half (51%) of the isolates were confirmed as *S.* typhimurium through amplification of the *fliC* gene (Table 3, Fig 2). Only a small proportion (32%) of the isolates was confirmed as *S. enteritidis* using the *Prot6e* gene PCR and the amplicons for representative isolates are shown in Fig 3. A small proportion (17%) of the confirmed *Salmonella* isolates were not positive for the *fliC* and *Prot6e* genes. The proportions of isolates positive for *Salmonella* species-specific and virulence genes did not vary (p > 0.05) with bird type.

### 3.2 Antimicrobial resistance profiles

A total of 82 confirmed *Salmonella* isolates were subjected to antimicrobial sensitivity test against a panel of 9 different antimicrobial agents (Table 4). Large proportions of the isolates were most often resistant to erythromycin (62%) and tetracycline (59%). On the contrary, smaller proportions of these isolates were resistant to trimethoprim (32%), amoxicillin (26%), ampicillin (22%), trimethoprim- sulfamethoxazole (18%) and kanamycin (15%). In addition, the isolates exhibited high susceptibility to chloramphenicol and gentamicin with 2% and 1% resistance recorded, respectively. Despite that aminoglycoside such as gentamicin appeared active *in vitro* against *Salmonella* isolates, it is not utilized clinically, hence it must be reported as resistant according to CLSI (2023 version) standards. The proportion of *Salmonella* isolates resistant to tested antibiotics were not influenced (p > 0.05) by the type of bird sampled.

Multiple antimicrobial resistance (MAR) phenotypes of isolates were generated (Table 5) using abbreviations on the antibiotic discs. All observed phenotypes were given a specific anti-biotypes codes (Ac) with a distinct number to differentiate between biotypes and they ranged between Ac1 and Ac20. Phenotypes Ac1 (16%), Ac2 (10%), Ac4 (10%), Ac10 (13%), and Ac19 (10%) were dominant across isolates. Isolate phenotypes Ac14 –Ac18 were resistant to five or more antibiotics. Phenotype Ac20 was resistant to the highest number (7) of antibiotics (ampicillin, chloramphenicol, erythromycin, trimethoprim-sulfamethoxazole, tetracycline, trimethoprim).

### 3.3 Detection of resistance genes

A total of 82 confirmed *Salmonella* isolates from different chicken types were subjected to PCR, targeting *ant (3”)-la*, *sul1*, *sul2*, *tet* (A), and *tet* (B) antimicrobial resistance genes (Table 6). Large proportions of the isolates possessed the *ant (3”)-la* (52%) and *Tet* (A) (46%) resistance genes. The proportions of isolates carrying *sul1* and *sul2* resistance genes ranged between 13% and 14% (see gene fragments in Figs 6 and 7, respectively). Agarose gel images of

**Table 3.  Proportions of isolates positive for *Salmonella* species-specific and virulence genes.**

| Bird type (%) | N | Percentage of isolates | | | Virulence genes | | |
| | | Genus and species-specific genes | | | | | |
| | | *InvA* | *fliC* | *prot6e* | *spiC* | *misL* | *orfL* |
|---|---|---|---|---|---|---|---|
| Broilers | 50 | 78 | 41 | 28 | 21 | 20 | 13 |
| Layers | 50 | 46 | 57 | 22 | 26 | 9 | 13 |
| Indigenous | 50 | 40 | 55 | 45 | 30 | 20 | 15 |
| Total | 150 | 55 | 49 | 30 | 26 | 16 | 14 |

Proportional data were analysed using the multinomial logistic regression procedure of SAS (2010)

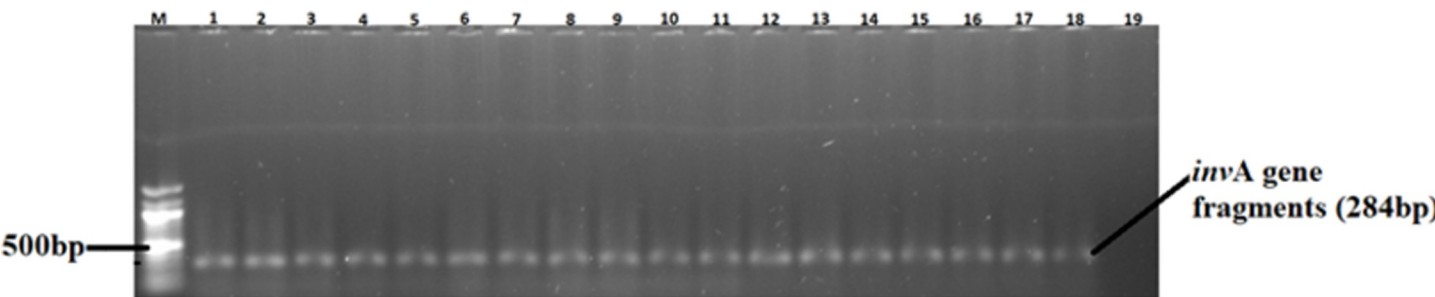

**Fig 1. A 1.5% (w/v) agarose gel image depicting _Salmonella_ species specific _invA_ gene from confirmed isolates.** Lane M = 100 bp DNA marker (Thermo Fisher Scientific, South Africa); Lane 1 = _invA_ gene fragment amplified from _Salmonella enteriditis_ positive control strain (ATCC: 13076TM); Lanes 2–18 = _Salmonella_ species specific _invA_ gene fragments amplified from the isolates; Lanes 19 = Negative control.

the _ant (3")-la_, _tet_ (A), _sul1_, and _sul2_ gene fragments are shown in Figs 4–7, respectively. On the other hand, only 9% of the isolates possessed _tet_ (B) resistant gene determinants. The proportion of isolates positive for antimicrobial resistance genes were not influenced (p > 0.05) by bird type.

### 3.4 Virulence genes in _Salmonella_ isolates

The 82 confirmed _Salmonella_ isolates were further screened for the presence of three virulence genes (_spiC_, _misL_, and _orfL_) using PCR. A moderate number (26%) of the isolates possessed the _spiC_ virulent gene (Table 3) whose gene fragments are shown in Fig 8. Only 16% of the isolates harboured the _misL_ virulent gene (Table 1) whose amplicons are shown in Fig 9. In addition, 14% of the isolates harboured the _orf_L virulent gene (see the gene fragments in Fig 10).

### 3.5 Phenotypic assessment of biofilm-formation

Only 69 of the 82 confirmed and preserved _Salmonella_ stock cultures were still viable after long-term storage at -80˚C. For this reason, 69 of the isolates were subjected to biofilm-formation analysis using microtiter plate assay. The results revealed that neither chicken type nor incubation temperature influenced biofilm-formation among the tested _Salmonella_ isolates. Based on the biofilm-formation patterns observed, isolates were classified as none, weak, moderate, or strong biofilm-forming strains. Regardless of incubation temperature, larger proportions of the isolates (35 to 62%) were categorized as strong biofilm-formers. The proportion of _Salmonella_ isolates that did not form biofilm ranged between 0 and 35%, while those that had moderate biofilm-forming capacity ranged between 0 and 20%. Between 4 and 35% of isolates

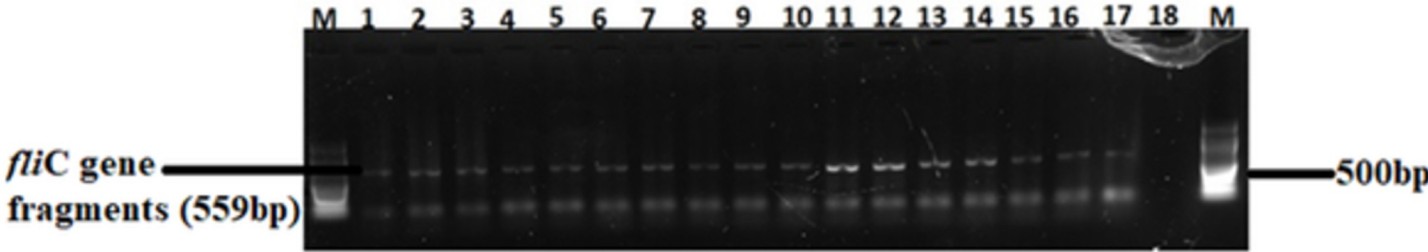

**Fig 2. A 1.5% (w/v) agarose gel image depicting _Salmonella_ species specific _fliC_ gene fragments from confirmed isolates.** Lane M = 100 bp DNA marker (Thermo Fisher Scientific, South Africa); Lane 11 = _fliC_ gene fragment amplified from _Salmonella typhimurium_ positive control strain (ATCC:14028TM); Lanes 1–10 and 12–17 = _Salmonella_ species specific _fliC_ gene fragments amplified from the isolates; Lane 18 = Negative control.

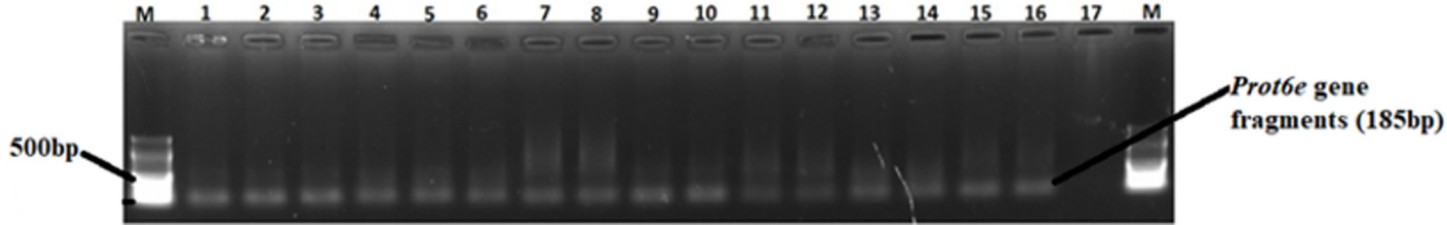

**Fig 3. A 1.5% (w/v) agarose gel image depicting *Salmonella* species specific *Prot6e* gene fragments amplified from confirmed isolates.** Lane M = 100 bp DNA marker (Thermo Fisher Scientific, South Africa); Lane 1 = *Prot6e* gene fragment amplified from *Salmonella enteriditis* positive control strain (ATCC: 13076TM); Lanes 2–16, *Salmonella* species specific *Prot6e* gene fragments amplified from the isolates; Lane 17 = Negative control.

were classified as weak biofilm-formers. The proportion of isolates that were able to form biofilms were not influenced (p > 0.05) by bird type.

## 4. Discussion

### 4.1 Prevalence of *Salmonella* in different poultry species

A total of 150 isolates were obtained from 15 layer, broiler, and dual-purpose indigenous chicken farms. *Salmonella* spp. identity was confirmed in 82 (55%) of the isolates by molecular detection of *invA* gene fragments. Similar results have been reported in previous studies [6, 38]. The occurrence of *Salmonella* spp. in poultry is influenced by several factors such as geographic location, prevention/control and biosafety measures of flocks, farm-specific husbandry practices, sampling season, and identification methods. In the current study, broilers had the highest *Salmonella* prevalence rate (78%), followed by layers (46%) and indigenous chickens (40%), as confirmed through PCR amplification of the *invA* gene. The observed variations in the occurrence of *Salmonella* spp. across broilers, layers and indigenous chickens may be attributed to the variation in management (biosecurity, hygiene, and sanitation) of the farms [39]. Despite that broilers and layers are raised under strict biosecurity standards compared to indigenous chickens, broilers had the highest occurrence rate of *Salmonella* spp. corroborating previous findings [40, 41]. *Salmonella enteritidis* and *S.* typhimurium are the most problematic zoonotic *Salmonella* serotypes that are responsible for serious human health infections globally [42]. In South Africa, they are also the most common serotypes reported in food producing animals including food products of animal origin [43, 44]. Confirmed *Salmonella* isolates were further identified using *Salmonella* species-specific gene fragments *fliC* (*S.* typhimurium) and *prot6e* (*S.* enteritidis). This analysis revealed that the prevalence of *S.* enteritidis and *S.* typhimurium serotypes was 49 and 30%, respectively. Only 21% of the confirmed *Salmonella*

**Table 4. Proportion of isolates resistant to tested antibiotics.** The superscript "a" indicate the percentage of *Salmonella* isolates that were resistant to the aminoglycosides (gentamicin and kanamycin) and these was reported based on the CLSI standards which stipulate that the above antimicrobial agents should not be reported as susceptible since they are not effective clinically.

| Bird type (%) | Percentage of isolates resistant to antibiotics | | | | | | | | |
|---|---|---|---|---|---|---|---|---|---|
| | N | AP30 | GM10[a] | K30[a] | C30 | A10 | E15 | TS25 | TM25 | T10 |
| Broilers | 39 | 13 | 100 | 100 | 0 | 18 | 46 | 13 | 36 | 49 |
| Layers | 23 | 30 | 100 | 100 | 4 | 26 | 78 | 17 | 22 | 74 |
| Indigenous | 20 | 30 | 100 | 100 | 5 | 40 | 75 | 30 | 35 | 60 |
| Total | 82 | 22 | 100 | 100 | 2 | 26 | 62 | 18 | 32 | 59 |

Ampicillin (AP30); Gentamicin (GM10); Kanamycin (K30); Chloramphenicol (C30); Amoxicillin (A10); Erythromycin (E15); Trimethoprim-sulfamethoxazole (TS25); Trimethoprim (TM25); Tetracycline (T10).

**Table 5. Antibiotic resistance phenotypes for *Salmonella* isolated from different chicken types.**

| Resistance phenotypes[1] | Proportion (%) | | | | Antibiotypes | Number of antibiotics | MAR Index |
|---|---|---|---|---|---|---|---|
| | Broiler | Layer | Indigenous | Total | | | |
| SXT-W-TE | 10 | 13 | 30 | 16 | Ac1 | 3 | 0.33 |
| E-W-TE | 13 | 9 | 5 | 10 | Ac2 | 3 | 0.33 |
| K-E-TE | 0 | 4 | 5 | 2 | Ac3 | 3 | 0.33 |
| AML-E-TE | 5 | 17 | 10 | 10 | Ac4 | 3 | 0.33 |
| AMP-AML-TE | 3 | 0 | 0 | 1 | Ac5 | 3 | 0.33 |
| AMP-C-AML | 0 | 4 | 0 | 1 | Ac6 | 3 | 0.33 |
| K-E-TE | 3 | 0 | 0 | 1 | Ac7 | 3 | 0.33 |
| AMP-SXT-W | 0 | 4 | 0 | 1 | Ac8 | 3 | 0.33 |
| CN-C-E | 0 | 4 | 0 | 1 | Ac9 | 3 | 0.33 |
| E-SXT-W-TE | 8 | 13 | 25 | 13 | Ac10 | 4 | 0.44 |
| AML-E-W-TE | 3 | 0 | 0 | 1 | Ac11 | 4 | 0.44 |
| AMP-AML-E-TE | 5 | 4 | 5 | 5 | Ac12 | 4 | 0.44 |
| K-AML.E-TE | 8 | 0 | 0 | 4 | Ac13 | 4 | 0.44 |
| AML-E-SXT-W-TE | 3 | 9 | 20 | 9 | Ac14 | 5 | 0.56 |
| AMP-AML-SXT-W-TE | 0 | 0 | 5 | 1 | Ac15 | 5 | 0.56 |
| AMP-AML-E-W-TE | 0 | 0 | 5 | 1 | Ac16 | 5 | 0.56 |
| AMP-K-AML.E-TE | 0 | 0 | 15 | 4 | Ac17 | 5 | 0.56 |
| K-E-SXT-W-TE | 3 | 0 | 0 | 1 | Ac18 | 5 | 0.56 |
| AMP-AML-E-SXT-W-TE | 0 | 17 | 20 | 10 | Ac19 | 6 | 0.67 |
| AMP-C-AML-E-SXT-W-TE | 0 | 4 | 0 | 1 | Ac20 | 7 | 0.78 |

[1]Phenotypes were generated using abbreviations that occur in the antibiotic discs.

N = 39, AMP, ampicillin; AML, amoxicillin; C, chloramphenicol; CN, gentamicin; E, erythromycin; K, kanamycin; STX, trimethoprim-sulfamethoxazole; TE, tetracycline; W, trimethoprim

isolates were negative for the two tested species-specific genes, suggesting that they could be other serotypes. The prevalence of *S*. typhimurium varied across layers (57%), broilers (41%), and indigenous chickens (55%). On the other hand, *S*. enteritidis was detected mostly in indigenous chickens (45%), followed by broilers (28%) and layers (22%). The lower prevalence of both *S*. typhimurium and *S*. enteritidis in broilers compared to layers and indigenous chickens can be attributed to the strict biosecurity measures employed for broiler production [14]. *Salmonella* can infect chickens without any clinical signs, leading to compromised productivity [45]. This highlights the importance of periodically screening chickens for the presence of *Salmonella* species to assess the risks and develop effective biosecurity measures to control their

**Table 6. Proportion of isolates positive for antimicrobial resistance genes.**

| Bird type (%) | N | *ant (3")-la* | Percentage of isolates positive for resistance genes | | | |
|---|---|---|---|---|---|---|
| | | | *sul1* | *sul2* | *tet* (A) | *tet* (B) |
| Broilers | 39 | 36 | 15 | 26 | 49 | 8 |
| Layers | 23 | 65 | 17 | 30 | 43 | 9 |
| Indigenous | 20 | 55 | 10 | 10 | 45 | 10 |
| Total | 82 | 52 | 14 | 13 | 46 | 9 |

Gentamicin (*ant (3")-la*); Tetracycline (*tet* (A) and *tet* (B)); Sulfamethoxazole (*sul1 and sul2*). Proportional data were analysed using the multinomial logistic regression procedure of SAS (2010)

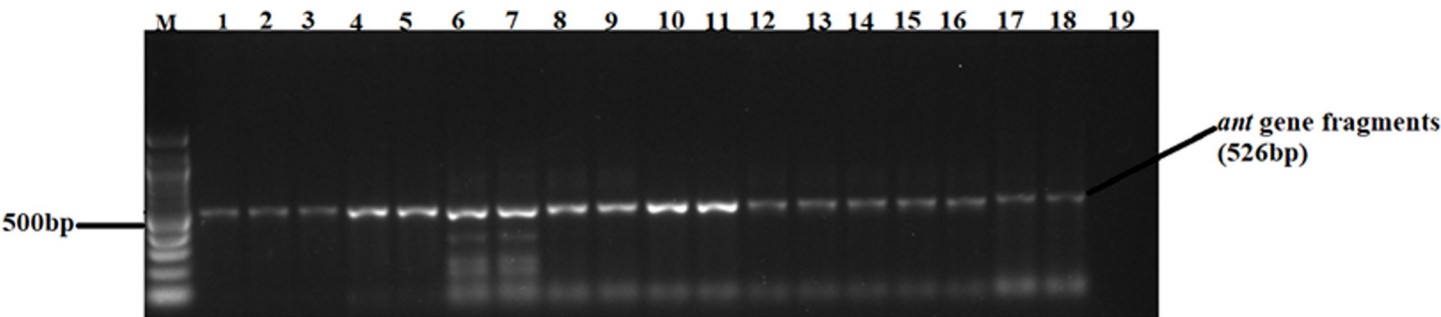

**Fig 4. A 1.5% (w/v) agarose gel image depicting the *ant (3")-la* resistance gene fragments from confirmed *Salmonella* isolates.** Lane M=100 bp DNA marker (Thermo Fisher Scientific, South Africa); Lanes 1–18, *ant (3")* resistance gene fragments amplified from the *Salmonella* isolates; Lane 19 = Negative control.

spread. Such interventions will reduce the incidence of *Salmonella* in poultry resulting in greater productivity and food safety as well as reduced public health concerns.

## 4.2 Antimicrobial resistance profiles of *Salmonella* isolates

Misuse or uncontrolled use of antibiotics in poultry production for growth promotion and prophylaxis has significantly contributed to the development of antimicrobial resistance among bacterial pathogens, including *Salmonella* [46]. Therefore, it is necessary to assess the antimicrobial resistance profile of isolates against commonly used antibiotics to develop more effective treatment and control strategies [47]. The isolates showed high resistance to erythromycin (62%), tetracycline (59%), and trimethoprim (32%). The observed high prevalence of resistance to certain drugs such as tetracyclines can be attributed to their common use in both animals and humans driven by affordability and accessibility, particularly in developing African countries [48, 49]. In comparison, a lower proportion of the isolates (15–26%) was resistant to amoxicillin, ampicillin, trimethoprim-sulfamethoxazole, and kanamycin, drugs that are not frequently used in the study area [30]. Other antibiotics such as ampicillin and amoxicillin are drugs of choice against Salmonellosis, hence the observed resistance may be attributed to their frequent use [50]. The lowest proportions of resistant isolates were observed for chloramphenicol (2%) and gentamicin (1%) antibiotics, reflecting the uncommon use of both drugs in the study area. Although. Aminoglycosides (gentamicin) may appear active *in vitro* against *Salmonella* spp. but are not effective clinically, hence it should not be reported as susceptible [32].

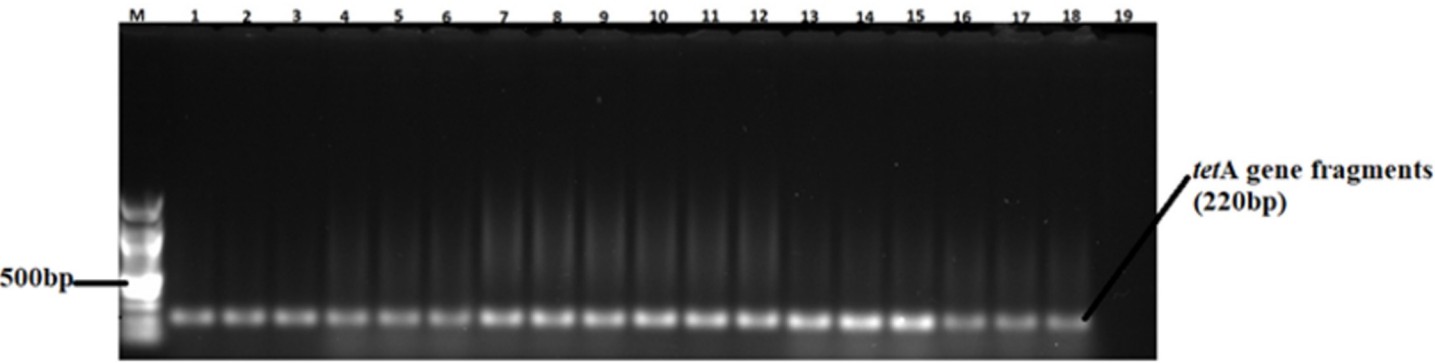

**Fig 5. A 1.5% (w/v) agarose gel image depicting the *tet* (A) resistance gene from confirmed *Salmonella* isolates.** Lane M=100 bp DNA marker (Thermo Fisher Scientific, South Africa); Lanes 1–18, *tet* (A) resistance gene fragments amplified from the isolates; Lane 19 = Negative control.

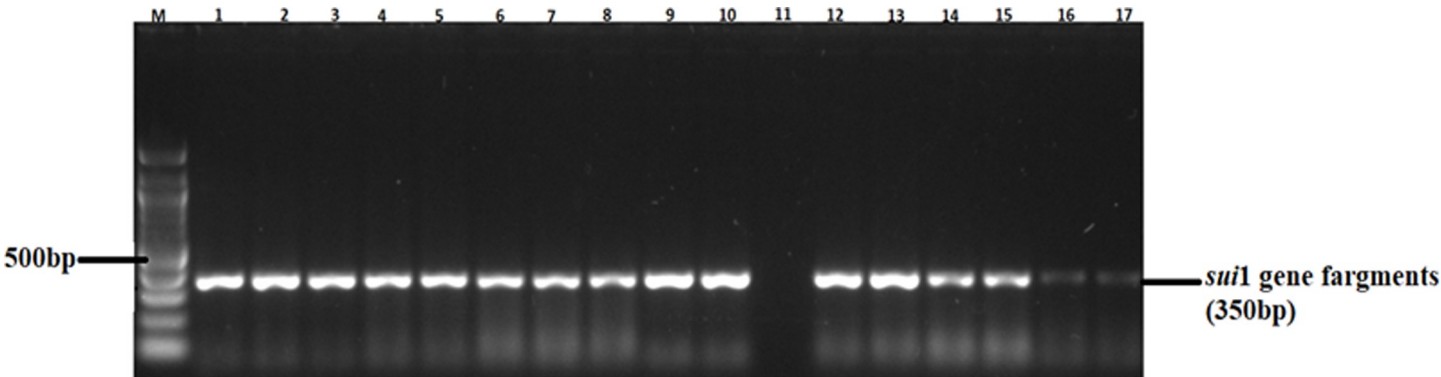

**Fig 6. A 1.5% (w/v) agarose gel image depicting the *sui*1 resistance gene fragments from confirmed *Salmonella* isolates.** Lane M=100 bp DNA marker (Thermo Fisher Scientific, South Africa); Lanes 1–10 and 12–17, *sui1* resistance gene fragments amplified from confirmed *Salmonella* isolates; Lane 11 = Negative control.

Interestingly, a high number of isolates from indigenous chickens showed resistance to several antibiotics and harboured resistance genes, suggesting potential of indirect transmission of antibiotic resistance. This phenomenon is more commonly reported in developing countries, particularly in poultry production settings characterized by limited resources and inadequate biosecurity measures [15, 18]. Additionally, bacterial pathogens disseminate resistance through horizontal gene transfer [51], and that may contribute to the spread of antimicrobial resistance to the environment and ultimately to indigenous chickens, particularly in areas with poor livestock waste management. In addition, the direct interaction or close proximity of the extensively reared indigenous chickens with other animals such as cattle, broilers, layers, and wild birds could have contributed to the observed findings.

Multiple drug resistance (MDR) is another growing global problem because it reduces disease treatment options [52]. *Salmonella* isolates assessed in the current study exhibited a high rate of MDR to three or more tested antibiotics classes with MAR index ranging from 0.33 to 0.78. Interestingly, the MAR index values were all above the 0.2 threshold, suggesting that the high-risk source of *Salmonella* pathogens contamination is where antimicrobial agents are frequently used [53]. This confirms that antibiotic use in poultry production farms does

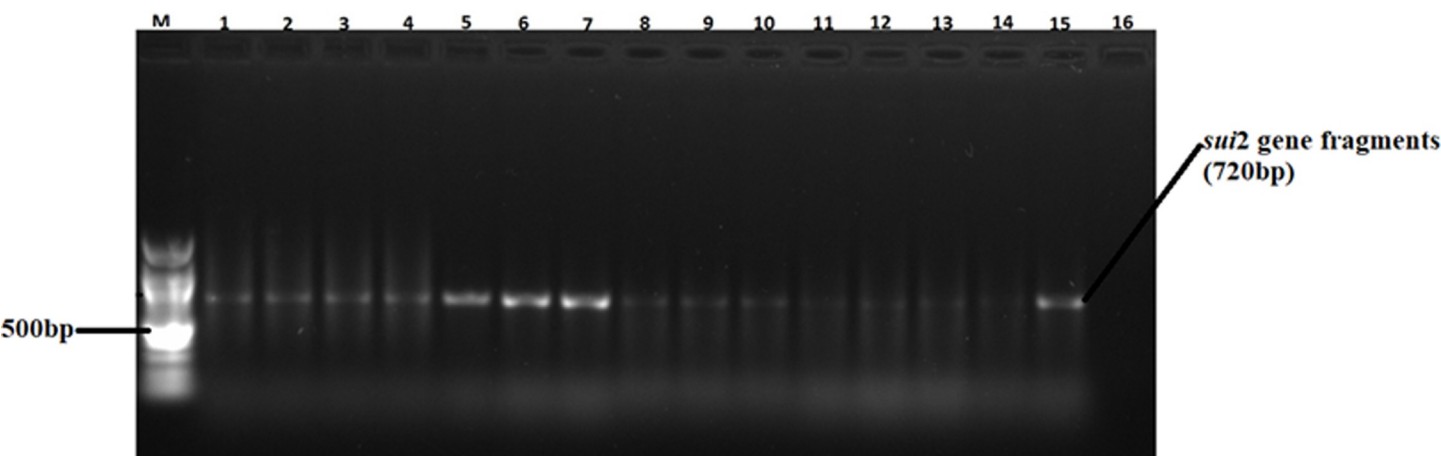

**Fig 7. A 1.5% (w/v) agarose gel image depicting the *sui2* resistance gene fragments from confirmed *Salmonella* isolates.** Lane M = 100 bp DNA marker (Thermo Fisher Scientific, South Africa); Lanes 1–15, *sui2* resistance gene fragments amplified from confirmed *Salmonella* isolates; Lane 16 = Negative control.

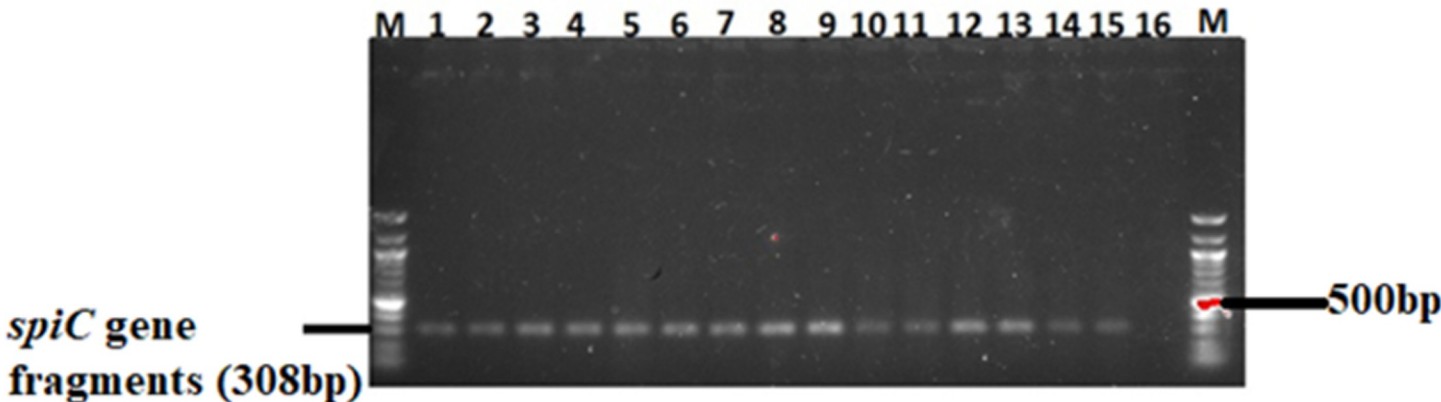

**Fig 8. A 1.5% (w/v) agarose gel image depicting the *spiC* virulence gene fragments from confirmed *Salmonella* isolates.** Lane M=100 bp DNA marker (Thermo Fisher Scientific, South Africa); Lanes 1–15, *Salmonella* species *spiC* virulent gene fragments; Lane 16 = Negative control.

contribute to the spread of resistant *Salmonella* pathogens. Several factors contribute to MDR development, including unregulated access to antibiotics and/or lack of compliance regarding the amount and type of antimicrobial agents used in poultry production and human medicine [54]. Some *Salmonella* isolates were resistant to more than five antibiotics, a major cause for concern given that diseases caused by such pathogens often have fatal outcomes [55]. Infections caused by MDR pathogens have severely limited treatment options thus putting animal and human lives at risks [38]. The adverse effects of MDR pathogens are a severe concern in developing countries due to inadequate health systems and limited resources to control them [55]. The detection of multiple resistant *Salmonella* strains in the three chicken types surveyed in this study has serious public health implications through food chain contamination [39].

### 4.3 Detection of resistance genes

The observed high rates of isolates AMR is not surprising, since most poultry producers in South Africa have unlimited over the counter access to most antimicrobials [56]. Among the 82 *Salmonella* isolates in the current study, 52, 46, 13, and 14% were positive for the *ant (3")-la*, *tet (A)*, *sui1*, and *sui2* resistant gene determinants, respectively. The results obtained from the analysis of resistance genes are consistent with the findings of the phenotype analysis,

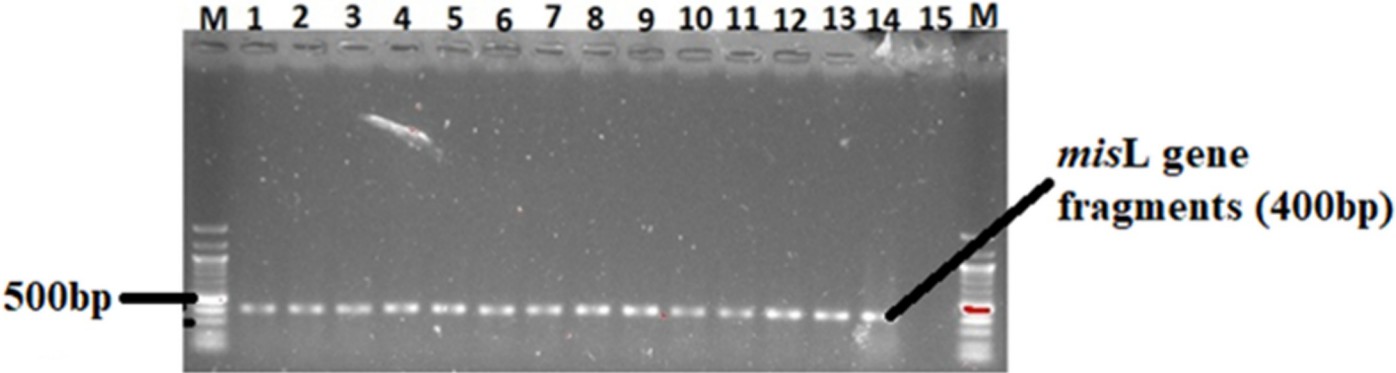

**Fig 9. A 1.5% (w/v) agarose gel image depicting the *misL* virulence gene from confirmed *Salmonella* isolates.** Lane M=100 bp DNA marker (Thermo Fisher Scientific, South Africa); Lanes 2–14 = *Salmonella* species *misL* virulent gene fragments from confirmed *Salmonella* isolates; Lane 15 = Negative control.

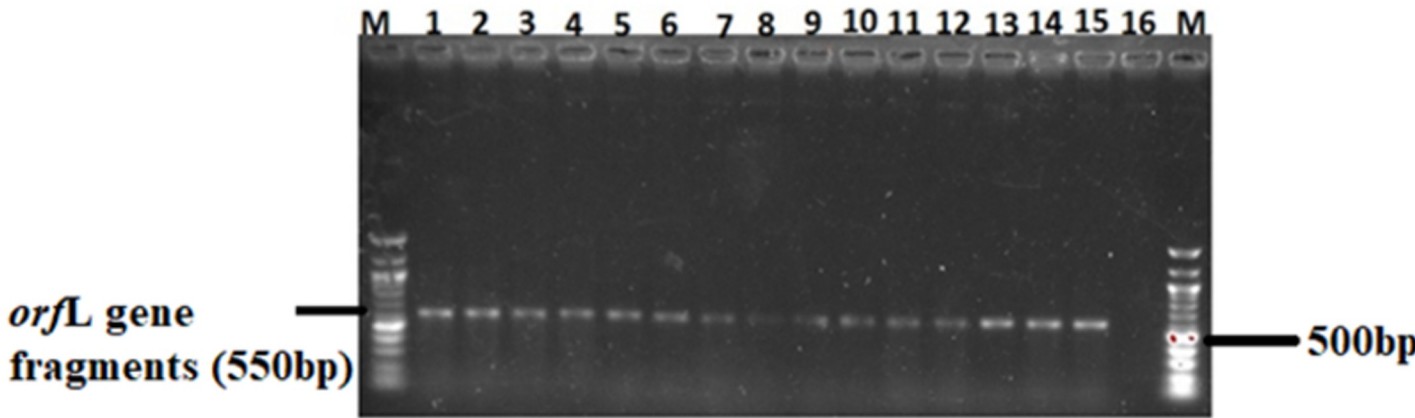

**Fig 10. A 1.5% (w/v) agarose gel image depicting the *orf*L virulence gene from confirmed *Salmonella* isolates.** Lane M = 100 bp DNA marker (Thermo Fisher Scientific, South Africa); Lanes 2–15 = *orf*L virulence gene fragments from confirmed *Salmonella* isolates; Lane 16 = Negative control.

especially in the case of tetracycline (59%) and trimethoprim-sulfamethoxazole (18%). Notably, there is a partial alignment between genotypic antibiotic resistance (AMR) and phenotypic antibiotic resistance results, suggesting the potential existence of silent antimicrobial resistance genes, particularly tet (A), tet (B) and *ant (3")-la* [57]. This phenomenon, also known as cryptic genes, has been observed in earlier studies [58–60]. The presence of silent antimicrobial resistance genes poses a new challenge in the battle against antimicrobial resistance, as it implies a risk not only with phenotypically resistant pathogens but also with antimicrobial-susceptible pathogens harbouring cryptic genes, as reported previously [61]. This dual risk emphasizes the complexity and potential covert nature of antimicrobial resistance. Despite the above, a small number of the isolates (9%) possessed the tet (B) resistance gene. Inexpensive and accessible antimicrobials such as tetracyclines tend to be abused in animal production and human medicine thus contributing to the development and spread of AMR [49].

Additionally, 56% of the isolates also harboured one of the aminoglycosides resistance genes called *ant (3")-la* gene. The above findings corroborate previous findings [62]. In the current study, the *ant (3")-la* gene was highly prevalent in layers (65%), followed by indigenous chickens (55%) and broilers (36%) among *Salmonella* isolates that had not shown phenotypic resistance to gentamicin. This suggests the potential presence of silent antimicrobial resistance genes among *Salmonella* isolates, posing a potential health risk for poultry producers in the study area. Although the *ant (3")-la* gene has been reported in several pathogenic bacterial strains, its prevalence in *Salmonella* isolates in Africa is not well-documented. In addition, genes harbouring resistance to sulfamethoxazole (*sul1* and *sul2*) were also detected in *Salmonella* isolates, including those resistant to trimethoprim-sulfamethoxazole. The occurrence of resistant *Salmonella* spp. suggests a need for alternatives to antibiotics [63], as well as treatment options in the event of disease outbreaks. Poultry production in low to medium income countries should be supported with alternative therapies against AMR pathogens, such as bacteriophage therapy.

## 4.4 Prevalence of virulence genes among *Salmonella* isolates

Virulence contributes to the invasiveness, pathogenicity, survival, and proliferation of *Salmonella* spp. [38]. Genetic determinants responsible for virulence help *Salmonella* to invade and destroy epithelial cells in host intestines and pave way for the colonization of other cell lines

[64]. In the current study, all three screened virulence genes belonged to different *Salmonella* pathogenicity islands, named SP1-2, SP1-3, and SP1-4 encoding for *spiC*, *misL*, and *orfL* genes, respectively. The most prevalent virulent gene was the *spiC*, found in 26% of the tested isolates, however, the role of *spiC* gene in the pathogenesis of *Salmonella* is yet to be unravelled [65]. The other virulent genes detected were the *misL* (16%) and the *orfL* (14%). Both genes have been associated with the survival of *Salmonella* in host macrophages during an infection [66]. The identification of virulent and AMR *Salmonella* pathogens raises concerns for both public health and the poultry industry. Antimicrobial-resistant pathogens not only lead to challenging-to-treat infections but also exacerbate infections and elevate the risks of mortality [67]. The detection of virulent genes in *Salmonella* strain at the farm level demonstrates the role played by healthy chickens in spreading pathogenic *Salmonella* strains to the environment or food chain leading to public health concerns [12, 68].

## 4.5 Phenotypic assessment of biofilm-formation

Biofilm-formation is a survival strategy used by pathogenic bacteria to evade harsh environments such as antibiotics and disinfectants while enhancing microorganisms' pathogenicity [69]. Bacterial biofilm-formation increases the burden of resistant pathogens and threatens food safety, especially when hygiene standards are compromised during food production and processing [70]. In the current study, isolates from different chicken types had similar biofilm-forming capacity. Most isolates were strong biofilm-formers, regardless of incubation temperature. These findings underscore the threat to food safety posed by the potential of *Salmonella* to form biofilms. Given that meat products are stored in cold facilities to mitigate foodborne poisoning incidents in humans, the observed robust biofilm formation by isolates, even at low temperatures (4˚C) typical of meat storage, raises significant concerns for food safety and public health. This heightened biofilm-forming nature at low temperatures increases the risk of meat contamination during storage, potentially leading to elevated morbidity and mortality cases, particularly among children and immunocompromised individuals [71]. This outcome necessitates the search for effective control strategies to ensure food safety and public health.

## 5. Conclusions

In conclusion, bird type or husbandry practices had no significant effects on the prevalence of AMR *Salmonella* spp. or resistance determinants. The detection of virulent pathogenic and AMR *Salmonella* spp. in the different chicken types suggest a public health risk and raises a concern for the South African poultry industry. This is because *Salmonella* is the most prevalent foodborne pathogen globally, frequently associated with the contamination of poultry products and diseases of economic and public health importance in poultry and humans. The occurrence of pathogenic and MDR *Salmonella* spp. in chickens suggests the need for careful evaluation of antibiotic use in all poultry production systems. Furthermore, it highlights the need to search for alternatives to prophylactic and therapeutic antibiotics such as bacteriophages.

## Supporting information

**S1 File. The survey questionare used for assessing poultry farms huasbandry practices.**
(DOCX)

**S2 File. Poultry farms husbandry practices and their characteristics.**
(DOCX)

**S1 Raw images. All uncropped and unadjusted gel images used in the manuscript.**
(PDF)

## Acknowledgments

The authors wish to thank Dr K.P. Montso for his support during the characterization analysis of *Salmonella* isolates.

## Author Contributions

**Conceptualization:** Sicelo B. Dlamini, Victor Mlambo, Caven Mguvane Mnisi, Collins Njie Ateba.

**Data curation:** Sicelo B. Dlamini, Victor Mlambo, Caven Mguvane Mnisi, Collins Njie Ateba.

**Formal analysis:** Sicelo B. Dlamini, Victor Mlambo, Caven Mguvane Mnisi, Collins Njie Ateba.

**Funding acquisition:** Sicelo B. Dlamini, Victor Mlambo, Caven Mguvane Mnisi, Collins Njie Ateba.

**Investigation:** Sicelo B. Dlamini, Victor Mlambo, Caven Mguvane Mnisi, Collins Njie Ateba.

**Methodology:** Sicelo B. Dlamini, Victor Mlambo, Caven Mguvane Mnisi, Collins Njie Ateba.

**Project administration:** Caven Mguvane Mnisi, Collins Njie Ateba.

**Resources:** Victor Mlambo, Caven Mguvane Mnisi, Collins Njie Ateba.

**Software:** Victor Mlambo.

**Supervision:** Victor Mlambo, Caven Mguvane Mnisi, Collins Njie Ateba.

**Validation:** Sicelo B. Dlamini, Victor Mlambo, Caven Mguvane Mnisi, Collins Njie Ateba.

**Visualization:** Sicelo B. Dlamini, Victor Mlambo, Caven Mguvane Mnisi, Collins Njie Ateba.

**Writing – original draft:** Sicelo B. Dlamini, Victor Mlambo, Caven Mguvane Mnisi, Collins Njie Ateba.

**Writing – review & editing:** Sicelo B. Dlamini, Victor Mlambo, Caven Mguvane Mnisi, Collins Njie Ateba.

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
