## [Decision Letter · Decision Letter 0]

20 Dec 2023

PONE-D-23-38462Virulence, multiple drug resistance, and biofilm formation in Salmonella species isolated from layer, broiler, and dual-purpose indigenous chickensPLOS ONE

Dear Dr. Dlamini,

Thank you for submitting your manuscript to PLOS ONE. After careful consideration, we feel that it has merit but does not fully meet PLOS ONE’s publication criteria as it currently stands. Therefore, we invite you to submit a revised version of the manuscript that addresses the points raised during the review process.

We look forward to receiving your revised manuscript.

Kind regards,

Raúl Alejandro Alegría-Morán, Ph.D.

Academic Editor

PLOS ONE

Journal Requirements:

3. We note that your Data Availability Statement is currently as follows: "All relevant data are within the manuscript and its Supporting Information files."

6. Please upload a copy of Supporting Information Figure/Table/etc. S1 Fig1, S2 Fig2, S3 Fig3, S4 Fig4, S5 Fig5, S6 Fig6, S7 Fig7, S8 Fig8, S9 Fig9, and S10 Fig10 which you refer to in your text on pages 31 and 32.

Reviewers' comments:

Reviewer's Responses to Questions

**Comments to the Author**

1. Is the manuscript technically sound, and do the data support the conclusions?

Reviewer #1: No

Reviewer #2: Yes

Reviewer #3: Partly

2. Has the statistical analysis been performed appropriately and rigorously? 

Reviewer #1: N/A

Reviewer #2: N/A

Reviewer #3: No

3. Have the authors made all data underlying the findings in their manuscript fully available?

Reviewer #1: Yes

Reviewer #2: Yes

Reviewer #3: Yes

4. Is the manuscript presented in an intelligible fashion and written in standard English?

Reviewer #1: No

Reviewer #2: Yes

Reviewer #3: Yes

5. Review Comments to the Author

Reviewer #1: With respect to the author's work and efforts, even if the study results would present importance for the scientific community in fighting against the antimicrobial resistance phenomenon, the study presentation modality and the used materials and methods section design do not meet the criterion for further processing in the prestigious PlosOne journal, mainly for the following reasons:

The manuscript contains numerous errors in grammar, punctuation, and sentence structure. The author may seek assistance from a scientific writer or revise the manuscript themselves, ensuring all necessary corrections are made before submission to another journal.

L33: “S. Typhimurium” – only “S.” must be italics, without to italicize Typhymurium

L53: “antimicrobials” instead of “antibiotics”, is a more appropriate term

L57 “enteritidis” – with sentence case

L87: the sampling strategy and design are unclear. How many samples were collected? The frequency of the collected samples and their amount are undefined. Was any sample size calculation taken into consideration? Information in farming systems are not described.

L116: the importance of the screening of InvA, fliC, and Prot6e genes (why the authors chose these genes) in molecular identification and confirmation of Salmonella isolates are not mentioned

L138: the authors not mentioned the using of positive controls within the PCR reactions in order to evidence the virulent genes. Therefore, the obtained results cannot be validated!

L150: the number of the tested antimicrobials is limited. It would be useful to present the classes they belong to

L166: the definition for multiple antibiotic resistance is wrong “multiple antibiotic resistance (MAR) phenotypes were generated for isolates that were resistant to three or more antimicrobial agents [26]”. Therefore, the accounted results are mistakenly presented. It is also unclear, why the authors used authors E. coli as positive control in testing antimicrobial susceptibility profile of Salmonella strains.

Reviewer #2: This study shows a hot issue that an important bacterial contaminated specie via the food chain such as Salmonella and also mentioned its antibiotic resistance regarding a public health concern. Although, the study is interesting, the manuscript is lacking some important information as follows:

Introduction:

-Page 4: lines 76-78: Mention Salmonella biofilm forming in a recent situation in poultry slaughter houses and prove it with reference (s). The biofilm-forming Salmonella and its antibiotic resistance status should also be mentioned.

Material and Methods:

-Page 4: line 88: “A total of 15 poultry farms…” Please provide total birds and the number of birds sampled from each farm.

Results:

-Page 4, lines 210-214: “Broiler chicken flocks (average size: 5000 birds)…, most of which had foot baths for biosecurity. Layer flocks... Indigenous chicken flocks… no biosecurity measures in place.” These are housing and hygienic background in each farm and chicken type. Information are not the results. These sentences should be moved to Materials and Methods section.

-Page 14, line 262: “Phenotype Ac1…Ac2…” Please, give the full name for Ac and then abbreviated.

Discussion

-Page 22-24: Discussion on more interesting topics should be added, including the relationship between phenotypic and genotypic antibiotic resistance (AMR), biofilm formation, and virulence genes discovered in Salmonella and their AMR characteristics.

-Page 15: As the result of MAR index values, discussion should be mentioned how important of the values are?

Reviewer #3: Abstract

Line 23-26: The Background statement should be collapsed into one sentence

Line 35-36: Not comprehensible. State the predominant patterns

Introduction

Line 80: Something is missing in “safety food products…”

The justification for the study seems insufficient. One is expected to understand whether studies on Salmonella in poultry have previously been conducted in the study area and the gap in knowledge

Line 81 and 84: biofilm forming should be one word “biofilm-forming”

Materials and Methods

Line 102 and 106: Check the correct way of writing Salmonella shigella agar

Line 136, 138, 140, 228, table 4: change virulent genes to virulence genes

Table 2: There is no need for the rows on mechanism of resistance and broad action

Line 152 -155: The choice of antibiotics for susceptibility testing should be based on standard recommendation by CLSI, EUCAST or any other organization. Testing the effect of an antibiotic against an organism that the drug is not indicated for its treatment, is not recommended because the need for the test is to guide therapy.

Line 159: recommended by which organization (CLSI, EUCAST?). There is need to follow standard procedure for reproducibility. The reference [23] in line 162 is not adequate in classifying the isolates as resistant or otherwise. It is recommended that CLSI in reference [26] criteria be used in classifying the isolates

Line 169-171, 430-432: Check the correct way of writing the resistance genes

Line 91-92: There is need for the questionnaire used in assessing poultry farm husbandry practices to be referenced as a supplementary material. The survey needs to be detailed; for example, how were the questions generated, what are the themes addressed by the questionnaire, how was the questionnaire pretested for reliability and validity, administered to avoid the farmers/respondents giving socially desirable answers, and the data analyzed. These are important for reproducibility.

Table 3: There may be no need for row on the mechanism of resistance

Line 182 to 183: The statement “Pseudomonas is a strong biofilm former” needs to be clarified. Does it mean that all Pseudomonas isolates/species are biofilm-producers? If no, for reproducibility, there is need to state the type culture of Pseudomonas strain used

Line 198 and Line 206: What is the meaning of SAS and LSMEANS, respectively?

Line 197-206: Statistical tool and data analysis only focused on optical density/biofilm formation. How were the other generated data analyzed? There would be need to test for statistical significance between the prevalence of Salmonella in the different chicken types since the prevalence results were compared/discussed extensively

Result

Line 219-220: Repetition of the methodology in line 117-118

Line 220-221: Recast as thus: “Out of 150 isolates, 82 (55%) were confirmed to be Salmonella”

Table 4: The title could be “Proportion of isolates positive for…”. Then the row with bird type should be “Bird type (%)” while the top will be “Percentage of isolates positive for specie-specific gene”. Moreover, are invA, fliC and prot6e virulence genes? If yes, why were they separated from others

Lie 254,411: gentamycin is incorrect

Table 5: The presentation is not comprehensible. For example, what does the figures represent? I assume it is the concentration of the antibiotics in the discs. Therefore, these concentrations should have been mentioned in the methodology. The tile should also be rephrased as mentioned earlier

Line 273-277: check the proper way of writing the genes

Table 7: There were two table 7 containing different information. The table 7 on biofilm is not necessary, it can be described

The gel pictures were not fully labelled. For example, which lane represents the marker and the isolates?

Discussion

Line 340-354, 397-403: Unnecessary

Line 356-363: Did not actually discuss the result of the study

Line 367-368: The comparison with references 6 and 36 seems inappropriate

Line 377: Did reference 38 and 39 compared Salmonella prevalence in the different chicken types?

Line 399: Italicize Salmonella

Line 40-402: This statement “In the current study, Salmonella isolates were tested against 9 different antibiotics and results revealed that” is unnecessary

Line 396-446: The resistance gens should be related to the phenotypes. If antimicrobial use was assessed by the questionnaire, it would have been easier to back-up the use of the antibiotics as the cause of the resistance. The authors detected resistance genes in isolates from indigenous chickens whereas antibiotics are not used in raising these birds. It is expected that the possible source(s) of the genes be postulated

Line 466-467: What is the implication of the finding that “Most isolates were strong biofilm formers, regardless of incubation temperature”

Conclusion

The conclusion did satisfactorily answer the objective of the study. For example, it did not capture the husbandry practices and prevalence/occurrence of Salmonella in the different production systems and the implication of finding resistance and virulence genes in Salmonella isolates from domesticated and indigenous poultry birds

The authors should state the limitations of the study

6. PLOS authors have the option to publish the peer review history of their article (what does this mean?). If published, this will include your full peer review and any attached files.

Reviewer #1: No

Reviewer #2: No

Reviewer #3: No

---

## [Author Response · Author response to Decision Letter 0]

2 Feb 2024

Reviewer #1: 

Comment: The manuscript contains numerous errors in grammar, punctuation, and sentence structure. The author may seek assistance from a scientific writer or revise the manuscript themselves, ensuring all necessary corrections are made before submission to another journal. Response: Thank you for the valuable comments. We attended to these language use issues as suggested.

Comment: L33: “S. Typhimurium” – only “S.” must be italics, without to italicize Typhimurium

Response: S. Typhimurium was changed to S. typhimurium

Comment: L53: “antimicrobials” instead of “antibiotics”, is a more appropriate term

Response: the term ‘antibiotics’ was changed to ‘antimicrobials’ as suggested. 

Comment: L57 “enteritidis” – with sentence case

Response: ‘enteritidis’ was changed as suggested.

Comment: L87: the sampling strategy and design are unclear. How many samples were collected? The frequency of the collected samples and their amount are undefined. Was any sample size calculation taken into consideration? Information in farming systems are not described. 

Response: The following information has been added: For microbiological analysis the minimum sample size of 75 was determined to be adequate for the study using a previously reported formular [22]. To achieve this, 5 samples were collected in each farm, directly from the cloaca of five randomly selected individual birds using sterile swabs containing multipurpose universal transport medium (line 127-129) 

The farming systems have now been described (Lines 119-126). 

Comment: L116: the importance of the screening of InvA, fliC, and Prot6e genes (why the authors chose these genes) in molecular identification and confirmation of Salmonella isolates are not mentioned

Response: The importance of screening the above indicated genes has been explained as follows: ‘This set of primers have been previously used to confirm the identity of the genus Salmonella and distinguish between S. typhimurium and S. enteriditis strains from other Salmonella serotypes [25]’. (Lines 162-165)

Comment: L138: the authors not mentioned the using of positive controls within the PCR reactions in order to evidence the virulent genes. Therefore, the obtained results cannot be validated! 

Response: The use of negative control eliminates false positive results. In addition, the electrophoresis gel pictures showed that all the amplicons reflected the expected size of the targeted genes. 

Comment: L150: the number of the tested antimicrobials is limited. It would be useful to present the classes they belong to

Response: The classes of antimicrobials to which the selected antibiotics belong have now been added (Lines 197-199).

Comment: L166: the definition for multiple antibiotic resistance is wrong “multiple antibiotic resistance (MAR) phenotypes were generated for isolates that were resistant to three or more antimicrobial agents [26]”. Therefore, the accounted results are mistakenly presented. 

Response: The definition has been revised to read: ‘Antibiotic resistance (MAR) phenotypes were generated for isolates that were resistant to two or more antimicrobial agents’ (Lines 209-211) 

Comment: It is also unclear, why the authors used authors E. coli as positive control in testing antimicrobial susceptibility profile of Salmonella strains. (All explained)

Response: Escherichia coli ATCC 25922 was used as a reference strain because it is a recommended strain for antimicrobial susceptibility test and its quality control guidelines permit greater accuracy in interpreting AMR results [29,30]. (Lines 207-209)

Reviewer #2: 

Comment: This study shows a hot issue that an important bacterial contaminated specie via the food chain such as Salmonella and also mentioned its antibiotic resistance regarding a public health concern. Although, the study is interesting, the manuscript is lacking some important information as follows:

Response: We are grateful for your valuable comments. We have responded to the comments below.

INTRODUCTION

Comment: -Page 4: lines 76-78: Mention Salmonella biofilm forming in a recent situation in poultry slaughter houses and prove it with reference (s). The biofilm-forming Salmonella and its antibiotic resistance status should also be mentioned.

Response: The following information has been provided: Multi-drug resistant Salmonella enterica with the ability to form biofilm has been reported in processing equipment’s in poultry facilities [20], suggesting increased risks for recurring contamination of poultry products. The potential of pathogenic bacteria including Salmonella species to form biofilms affects food safety even for products preserved in appropriate storage and refrigerated temperatures [21] (Lines 88-93)

MATERIAL AND METHODS

Comment: Page 4: line 88: “A total of 15 poultry farms…” Please provide total birds and the number of birds sampled from each farm.

Response: The following information has been provided: A total of 75 faecal samples were collected from 15 poultry farms (5 layers, 5 broilers, and 5 indigenous dual-purpose chickens). The faecal samples were collected directly from the cloaca of five randomly selected individual birds per farm, using sterile swabs containing multipurpose universal transport medium. (Lines 131-135) 

RESULTS

Comment:-Page 4, lines 210-214: “Broiler chicken flocks (average size: 5000 birds)…, most of which had foot baths for biosecurity. Layer flocks... Indigenous chicken flocks… no biosecurity measures in place.” These are housing and hygienic background in each farm and chicken type. Information are not the results. These sentences should be moved to Materials and Methods section. 

Response: The correction has been done as suggested. (line 119-126)

Comment: Page 14, line 262: “Phenotype Ac1…Ac2…” Please, give the full name for Ac and then abbreviated.

Responses: All observed phenotypes were given a specific antibiotypes codes (Ac) with a distinct number to differentiate between biotypes and they ranged between Ac1 and Ac20 (Lines 302-305)

DISCUSSION

Comment: Page 22-24: Discussion on more interesting topics should be added, including the relationship between phenotypic and genotypic antibiotic resistance (AMR), biofilm formation, and virulence genes discovered in Salmonella and their AMR characteristics.

Response: Discussions on the above suggested topics have been added. Lines 461 – 471 and Lines 516 – 524.

Comment: Page 15: As the result of MAR index values, discussion should be mentioned how important of the values are?

Response: The discussion on the importance of the MAR index values has been included (line 444-447) 

Reviewer #3: 

Abstract

Comment: Line 23-26: The Background statement should be collapsed into one sentence

Response: The background statement has now been condensed into one sentence as follows: ‘Globally, the significant risk to food safety and public health posed by antimicrobial-resistant foodborne Salmonella pathogens is driven by the utilization of in-feed antibiotics, with variations in usage across poultry production systems.’ (line 23-25)

Comment: Line 35-36: Not comprehensible. State the predominant patterns 

Response: The dominant antibiotic resistance phenotypes were: SXT-W-TE (16%), E-W-TE (10%), AML-E-TE (10%), E-SXT-W-TE (13%), and AMP-AML-E-SXT-W-TE (10%). (line 34-36). 

INTRODUCTION

Comment: Line 80: Something is missing in “safety food products…”

Response: Revised to ‘safety of food products’ 

Comment: The justification for the study seems insufficient. One is expected to understand whether studies on Salmonella in poultry have previously been conducted in the study area and the gap in knowledge

Response: The following has been added to the justification section: ‘Several studies conducted on Salmonella species in poultry from study area focused on the prevalence of virulent and AMR Salmonella serovars [10;11;12], with no consideration of the influence of poultry production systems and management practices.’ (line 71 – 73)

Comment: Line 81 and 84: biofilm forming should be one word “biofilm-forming”

Response: Revised accordingly. 

MATERIALS AND METHODS

Comment: Line 102 and 106: Check the correct way of writing Salmonella shigella agar

Response: ‘Salmonella shigella agar’ was revised to ‘Salmonella Shigella Agar (SSA)’

Comment: Line 136, 138, 140, 228, table 4: change virulent genes to virulence genes

Response: ‘Virulent genes’ was revised to ‘virulence genes’

Comment: Table 2: There is no need for the rows on mechanism of resistance and broad action –

Response: Rows on mechanism of resistance and broad action in Table 2 was removed. 

Comment: Line 152 -155: The choice of antibiotics for susceptibility testing should be based on standard recommendation by CLSI, EUCAST or any other organization. Testing the effect of an antibiotic against an organism that the drug is not indicated for its treatment, is not recommended because the need for the test is to guide therapy.

Response: The statement was revised as follows: ‘The choice of selected antibiotics was based on the standard recommendation by CLSI, particularly antibiotics that are commonly used in the treatment of bacterial infections in both humans and animals.’ (line 191-193)

Comment: Line 159: recommended by which organization (CLSI, EUCAST?). There is need to follow standard procedure for reproducibility. The reference [23] in line 162 is not adequate in classifying the isolates as resistant or otherwise. It is recommended that CLSI in reference [26] criteria be used in classifying the isolates

Response: The sentence has been revised to indicate that the recommendation was by CLSI organization. The CLSI in reference [30] (previously reference [26]) criteria was used to classify the isolates. (line 205-207)

Comment: Line 169-171, 430-432: Check the correct way of writing the resistance genes 

Response: The resistance genes were revised to sul1 and sul2

Comment: Line 91-92: There is need for the questionnaire used in assessing poultry farm husbandry practices to be referenced as a supplementary material. The survey needs to be detailed; for example, how were the questions generated, what are the themes addressed by the questionnaire, how was the questionnaire pretested for reliability and validity, administered to avoid the farmers/respondents giving socially desirable answers, and the data analyzed. These are important for reproducibility.

Response: The questionnaire has now been referenced as a supplementary material. (line 107-108)

To provide additional detail regarding the survey, the following information has been provided: ‘Prior to sample collection 15 (5 broilers, 5 layers and 5 dual-purpose indigenous chickens) poultry farms were identified and selected. The owners of the farms were approached to participate in the study based on willingness. Data on antibiotic treatment history and related husbandry practices were collected from 15 poultry farms (5 broilers, 5 layers and 5 dual-purpose indigenous chickens) through a structured questionnaire (S1 Survey). The questionnaire elicited information related to the poultry farmers’ demographic, husbandry practices, antibiotic use, and their knowledge of Salmonella spp. The survey instrument was face and content validated prior to administration by experts in the field of study. The reliability test of the instrument was carried out through the test-re-test reliability procedure by administering the questionnaire to two (2) poultry farmers at an interval of one week. The responses from the two administrations were then correlated and a high correlation coefficient of r = 0.80 was obtained. This confirmed the consistency and reliability of the instrument. Furthermore, a multiple contact strategy was used to eliminate sampling error and to ensure accuracy of responses gathered from the farmers. These eliminated the risks of receiving socially desirable responses from the farmers.’ (line 102-118) 

Comment: Table 3: There may be no need for row on the mechanism of resistance

Response: The row on the mechanism of resistance was removed from Table 3 which is now table 2

Comment: Line 182 to 183: The statement “Pseudomonas is a strong biofilm former” needs to be clarified. Does it mean that all Pseudomonas isolates/species are biofilm-producers? If no, for reproducibility, there is need to state the type culture of Pseudomonas strain used

Response: We have now indicated that it is Pseudomonas aeruginosa ATCC 27853 that is being referred to. 

Comment: Line 198 and Line 206: What is the meaning of SAS and LSMEANS, respectively?

Responses: These acronyms have now been written in full as: Statistical Analysis System (SAS) and Least squares means (LSMEANS) 

Comment: Line 197-206: Statistical tool and data analysis only focused on optical density/biofilm formation. How were the other generated data analyzed? There would be need to test for statistical significance between the prevalence of Salmonella in the different chicken types since the prevalence results were compared/discussed extensively

Response: Agreed. We have now analysed the remaining data and added the following information to the revised manuscript: ‘Proportional data (arising from discrete counts) were analysed using the multinomial logistic regression procedure of SAS (2010). In the categorical variable 'bird type,' the reference category selected was broiler due to the reported extensive use of antibiotic growth promoters in this group. Consequently, there was an anticipation of a higher likelihood of detecting antimicrobial resistance in broilers.’(line 252-256)

RESULT

Comment: Line 219-220: Repetition of the methodology in line 117-118

Responses: The repetition was removed.

Comment: Line 220-221: Recast as thus: “Out of 150 isolates, 82 (55%) were confirmed to be Salmonella”

Responses: Revised as suggested. (line 259)

Comment: Table 4: The title could be “Proportion of isolates positive for…”. Then the row with bird type should be “Bird type (%)” while the top will be “Percentage of isolates positive for specie-specific gene”. 

Responses: Thank you for these suggestions. We have modified the table caption accordingly.

Comment: Moreover, are invA, fliC and prot6e virulence genes? If yes, why were they separated from others- 

Responses: Thank you for this comment; The genes are virulent genes, particularly invA gene, hence the tables have now been merged into one. Despite the above, virulent genes are sometimes used as biomarkers to confirm the identity of isolates or Salmonella serovars. This is because the invA gene contains sequences unique to the genus Salmonella while fliC and prot6e contain sequences unique to S. typhimurium and S. enteritidis, respectively and they have been previously used as biomarkers for Salmonella genus and serovars [25]. 

Comment: Lie 254,411: gentamycin is incorrect

Response: ‘gentamycin’ was corrected to ‘gentamicin’

Comment: Table 5: The presentation is not comprehensible. For example, what does the figures represent? I assume it is the concentration of the antibiotics in the discs. Therefore, these concentrations should have been mentioned in the methodology. 

Response: The concentration of the antibiotics were mentioned in the methodology as follows: The list of used antibiotics comprised of gentamicin (GM10 μg), amoxicillin (A10 μg), erythromycin (E15 μg), chloramphenicol (C30 μg), tetracycline (T10 μg), trimethoprim (TM25 μg), ampicillin (AP30 μg), trimethoprim-sulfamethoxazole (TS25 μg), and kanamycin (K30 μg) [26, ] (line 193-197)

Comment: The title should also be rephrased as mentioned earlier 

Response: The title has now been revised to read: Proportion of isolates resistant to tested antibiotics.

Comment: Line 273-277: check the proper way of writing the genes

Response: All genes were corrected as follows: ‘ant (3”)-la, sui1, sui2, tet (A), and tet (B)’

Comment: Table 7: There were two table 7 containing different information. The table 7 on biofilm is not necessary, it can be described 

Response: The table 7 on biofilm has been removed and results described as suggested. (line 369-379)

Comment: The gel pictures were not fully labelled. For example, which lane represent

---

## [Decision Letter · Decision Letter 1]

28 Feb 2024

PONE-D-23-38462R1Virulence, multiple drug resistance, and biofilm-formation in Salmonella species isolated from layer, broiler, and dual-purpose indigenous chickensPLOS ONE

Dear Dr. Dlamini,

Thank you for submitting your manuscript to PLOS ONE. After careful consideration, we feel that it has merit but does not fully meet PLOS ONE’s publication criteria as it currently stands. Therefore, we invite you to submit a revised version of the manuscript that addresses the points raised during the review process. Please consider particularly what is exposed by reviewers 1 and 2 for the new version of the manuscript. 

We look forward to receiving your revised manuscript.

Kind regards,

Raúl Alejandro Alegría-Morán, Ph.D.

Academic Editor

PLOS ONE

Journal Requirements:

Reviewers' comments:

Reviewer's Responses to Questions

**Comments to the Author**

1. If the authors have adequately addressed your comments raised in a previous round of review and you feel that this manuscript is now acceptable for publication, you may indicate that here to bypass the “Comments to the Author” section, enter your conflict of interest statement in the “Confidential to Editor” section, and submit your "Accept" recommendation.

Reviewer #1: (No Response)

Reviewer #2: (No Response)

Reviewer #3: All comments have been addressed

2. Is the manuscript technically sound, and do the data support the conclusions?

Reviewer #1: No

Reviewer #2: Partly

Reviewer #3: Yes

3. Has the statistical analysis been performed appropriately and rigorously? 

Reviewer #1: N/A

Reviewer #2: N/A

Reviewer #3: Yes

4. Have the authors made all data underlying the findings in their manuscript fully available?

Reviewer #1: Yes

Reviewer #2: Yes

Reviewer #3: Yes

5. Is the manuscript presented in an intelligible fashion and written in standard English?

Reviewer #1: Yes

Reviewer #2: Yes

Reviewer #3: Yes

6. Review Comments to the Author

Reviewer #1: Even though the authors tried to address the concerns raised by the reviewer, the wording below, in response to a critical concern, compromises the results of the study.

Previous reviewer comment:

Comment: L166: the definition for multiple antibiotic resistance is wrong “multiple

antibiotic resistance (MAR) phenotypes were generated for isolates that were resistant

to three or more antimicrobial agents [26]”. Therefore, the accounted results are

mistakenly presented.

Author response in revision

"Response: The definition has been revised to read: ‘Antibiotic resistance (MAR)

phenotypes were generated for isolates that were resistant to two or more antimicrobial

agents’ (Lines 209-211)"

This statement is scientifically invalid. Please consult: doi: 10.1111/j.1469-0691.2011.03570.x.

Magiorakos et al., 2012

Reviewer #2: Revision has improved and provided more important aspects of the manuscript. However, some concerns are as follows:

Introduction:

-Page 4: lines 92: Prevalence of biofilm-forming Salmonella and its antimicrobial resistance in poultry farms in your country or other countries in previous or recently studies should also be mentioned.

Discussion

-Page 23/ line 440: The sentence, “…MDR to more than two tested antibiotics…” should be changed to “…MDR to more than two tested antibiotic classes…” because MDR means bacteria resisted to more than two tested antibiotic classes not only resisted to the antibiotic types.

Other

Table 2: “…of resistant genes in Salmonella isolates”. The phrase is not complete, should be changed to “…of resistant antimicrobial genes in Salmonella isolates”.

Table 3,4,6: The data (genus, specie, and virulence genes or resistant isolates) have presented as the percentage. However, the number of resulted isolates are not clear, so the authors should provide the number of isolates (maybe in parenthesis) for each observed value.

Table 3,4,6: Please, express the statistical analysis in the tables along with the table legends.

All figures (gel pictures): The author should provide the company and country of DNA maker used.

Reviewer #3: The authors diligently addressed my comments and there was no detection of dual publication or ethical concern.

7. PLOS authors have the option to publish the peer review history of their article (what does this mean?). If published, this will include your full peer review and any attached files.

Reviewer #1: No

Reviewer #2: No

Reviewer #3: No

---

## [Author Response · Author response to Decision Letter 1]

26 Mar 2024

Reviewer #1: Even though the authors tried to address the concerns raised by the reviewer, the wording below, in response to a critical concern, compromises the results of the study.

Previous reviewer comment:

Comment: L166: the definition for multiple antibiotic resistance is wrong “multiple

antibiotic resistance (MAR) phenotypes were generated for isolates that were resistant

to three or more antimicrobial agents [26]”. Therefore, the accounted results are

mistakenly presented.

Author response in revision

"Response: The definition has been revised to read: ‘Antibiotic resistance (MAR)

phenotypes were generated for isolates that were resistant to two or more antimicrobial

agents’ (Lines 209-211)"

This statement is scientifically invalid. Please consult: doi: 10.1111/j.1469-0691.2011.03570.x.

Magiorakos et al., 2012.

Response: Thank you for this comment, the statement has been now revised to: “multiple antibiotic resistance (MAR) phenotypes were generated for isolates that were resistant to at least one agent in three or more antimicrobial categories’’. [Lines 206-209]

Reviewer #2: Revision has improved and provided more important aspects of the manuscript. However, some concerns are as follows:

Introduction:

-Page 4: lines 92: Prevalence of biofilm-forming Salmonella and its antimicrobial resistance in poultry farms in your country or other countries in previous or recently studies should also be mentioned.

Response: Revised as follows: Recent studies have reported a high prevalence of multi-drug resistant biofilm-forming Salmonella serovars in poultry farms and processing facilities globally [20,21,22,23,24], suggesting increased risks for recurring contamination of poultry products. [Lines 88-91]

Discussion

Comment

-Page 23/ line 440: The sentence, “…MDR to more than two tested antibiotics…” should be changed to “…MDR to more than two tested antibiotic classes…” because MDR means bacteria resisted to more than two tested antibiotic classes not only resisted to the antibiotic types.

Response: Thank you for this comment and the statement has been revised to: “MDR to three or more tested antibiotics classes”. [Line 454]

Other 

Comments 

Table 2: “…of resistant genes in Salmonella isolates”. The phrase is not complete, should be changed to “…of resistant antimicrobial genes in Salmonella isolates”.

Response: The phrase has been revised to: “of resistant antimicrobial genes in Salmonella isolates”. [Table 2]

Comment

Table 3,4,6: The data (genus, specie, and virulence genes or resistant isolates) have presented as the percentage. However, the number of resulted isolates are not clear, so the authors should provide the number of isolates (maybe in parenthesis) for each observed value.

Response: Thank you for the comment, the total number of isolates (N) are indicated in table 3,4,6, to make it easier for readers to calculate the number of isolates without cluttering the tables by presenting both the number and percentage in each cell. 

Comment

Table 3,4,6: Please, express the statistical analysis in the tables along with the table legends.

Response: The statistical analysis has now been stated in the table footnotes (Table 3,4,6) as follows: “Proportional data were analysed using the multinomial logistic regression procedure of SAS (2010)” 

Comment

All figures (gel pictures): The author should provide the company and country of DNA maker used.

Response: The company and country of DNA maker used have now been provided in all gel picture legends as follows: “Thermo Fisher Scientific, South Africa”

---

## [Decision Letter · Decision Letter 2]

8 May 2024

PONE-D-23-38462R2Virulence, multiple drug resistance, and biofilm-formation in Salmonella species isolated from layer, broiler, and dual-purpose indigenous chickensPLOS ONE

Dear Dr. Dlamini,

Thank you for submitting your manuscript to PLOS ONE. After careful consideration, we feel that it has merit but does not fully meet PLOS ONE’s publication criteria as it currently stands. Therefore, we invite you to submit a revised version of the manuscript that addresses the points raised during the review process. **Please take into consideration the comments and suggestions of reviewer 3, which are necessary for the final acceptance of the manuscript.**

We look forward to receiving your revised manuscript.

Kind regards,

Raúl Alejandro Alegría-Morán, Ph.D.

Academic Editor

PLOS ONE

Journal Requirements:

Reviewers' comments:

Reviewer's Responses to Questions

**Comments to the Author**

1. If the authors have adequately addressed your comments raised in a previous round of review and you feel that this manuscript is now acceptable for publication, you may indicate that here to bypass the “Comments to the Author” section, enter your conflict of interest statement in the “Confidential to Editor” section, and submit your "Accept" recommendation.

Reviewer #3: All comments have been addressed

2. Is the manuscript technically sound, and do the data support the conclusions?

Reviewer #3: Yes

3. Has the statistical analysis been performed appropriately and rigorously? 

Reviewer #3: I Don't Know

4. Have the authors made all data underlying the findings in their manuscript fully available?

Reviewer #3: Yes

5. Is the manuscript presented in an intelligible fashion and written in standard English?

Reviewer #3: Yes

6. Review Comments to the Author

**Reviewer #3:** 39: Give space theSPI-3

102: Put a comma after “Prior to sample collection”

141: Write “Salmonella Shigella Agar” as Salmonella-Shigella agar and italicize Salmonella and Shigella

142: Change “Lactose-negative” to Lactose non-fermenting

144: Change “lactose-positive” to “lactose-fermenting”

305: This statement should not be under table 4 as the table did not show any data analyzed with the tool “Proportional data were analysed using the multinomial logistic regression procedure of SAS (2010)” Further, it could be observed that all the isolates from broilers and layers were reported as susceptible to gentamicin – this report is contrary to the cited CLSI document which stated that Salmonella isolates should always be reported as resistant to gentamicin and streptomycin because these aminoglycosides although might show in vitro activity but are not utilized clinically

320-325: Correct “sui1, sui2,” They should be sul1, sul2 and italicized. Also correct them in line 494 and table 6

483: Check tet (B)

S, streptomycin appeared as a legend underneath table 5 whereas there was no resistance to streptomycin

475-476: Give examples of the silent genes observed in your experiment

486: resistance not resistant

533: Put a comma after “In conclusion”

7. PLOS authors have the option to publish the peer review history of their article (what does this mean?). If published, this will include your full peer review and any attached files.

Reviewer #3: No

---

## [Author Response · Author response to Decision Letter 2]

18 Jun 2024

Reviewer #3: 

Comment: 39: Give space theSPI-3

Response: theSPI-3 was revised to ‘the SPI-3’

Comment: 102: Put a comma after “Prior to sample collection”

Response: Done as suggested.

Comment: 141: Write “Salmonella Shigella Agar” as Salmonella-Shigella agar and italicize Salmonella and Shigella

Response: “Salmonella Shigella Agar” was revised as advised to ‘Salmonella-Shigella agar’ 

Comment: 142: Change “Lactose-negative” to Lactose non-fermenting

Response: “Lactose-negative” was revised to ‘Lactose non-fermenting’

Comment: 144: Change “lactose-positive” to “lactose-fermenting”

Response: “lactose-positive” was revised to ‘lactose-fermenting’ 

Comment: 305: This statement should not be under table 4 as the table did not show any data analyzed with the tool “Proportional data were analysed using the multinomial logistic regression procedure of SAS (2010)” Further, it could be observed that all the isolates from broilers and layers were reported as susceptible to gentamicin – this report is contrary to the cited CLSI document which stated that Salmonella isolates should always be reported as resistant to gentamicin and streptomycin because these aminoglycosides although might show in vitro activity but are not utilized clinically.

Response: Thank you for this comment, the statement was removed under table 4. Thank you for this observation and we do agree with you. To clarify the above, the following statement was added: “Despite that aminoglycosides such as gentamicin and kanamycin displayed in-vitro activity against the Salmonella isolates in this study, these isolates should be considered as resistant based on the CLSI recommendation, since these drugs are not used clinically.” However, there is evidence of a report indicating that Salmonella infections caused by isolates that are resistant to first line antibiotics may be treated with alternative second line antimicrobial agents such as aminoglycosides principally amikacin, or folic acid pathway inhibitors like sulfisoxazole or sulfamethoxazole with or without trimethoprim as well as carbapenems, imipenem or meropenem which are administered intravenously (Guerrant et al., 2001). 

Comment: 320-325: Correct “sui1, sui2,” They should be sul1, sul2 and italicized. Also correct them in line 494 and table 6

Response: Done as suggested.

Comment: 483: Check tet (B)

S, streptomycin appeared as a legend underneath table 5 whereas there was no resistance to streptomycin

Response: S, streptomycin was removed as a legend underneath table 5 and tet (B) was fixed 

Comment: 475-476: Give examples of the silent genes observed in your experiment

Response: Thank you for this comment and the silent genes were added as follows; particularly tet (A), tet (B) and ant (3”)-la

Comment: 486: resistance not resistant

Response: ‘Resistant’ was revised to ‘resistance’ 

Comment: 533: Put a comma after “In conclusion”

Response: Done as suggested.

---

## [Decision Letter · Decision Letter 3]

23 Aug 2024

Virulence, multiple drug resistance, and biofilm-formation in Salmonella species isolated from layer, broiler, and dual-purpose indigenous chickens

PONE-D-23-38462R3

Dear Dr. Dlamini,

We’re pleased to inform you that your manuscript has been judged scientifically suitable for publication and will be formally accepted for publication once it meets all outstanding technical requirements.

Kind regards,

Raúl Alejandro Alegría-Morán, Ph.D.

Academic Editor

PLOS ONE

Additional Editor Comments (optional):

Reviewers' comments:

Reviewer's Responses to Questions

**Comments to the Author**

1. If the authors have adequately addressed your comments raised in a previous round of review and you feel that this manuscript is now acceptable for publication, you may indicate that here to bypass the “Comments to the Author” section, enter your conflict of interest statement in the “Confidential to Editor” section, and submit your "Accept" recommendation.

Reviewer #3: All comments have been addressed

2. Is the manuscript technically sound, and do the data support the conclusions?

Reviewer #3: Yes

3. Has the statistical analysis been performed appropriately and rigorously? 

Reviewer #3: Yes

4. Have the authors made all data underlying the findings in their manuscript fully available?

Reviewer #3: Yes

5. Is the manuscript presented in an intelligible fashion and written in standard English?

Reviewer #3: Yes

6. Review Comments to the Author

Reviewer #3: Comments have been addressed. All questions have been answered ethics in research were adhered to and formatting style is in accordance with the Journal's guidelines

7. PLOS authors have the option to publish the peer review history of their article (what does this mean?). If published, this will include your full peer review and any attached files.

Reviewer #3: No

---

## [Editor Report · Acceptance letter]

28 Aug 2024

PONE-D-23-38462R3 

PLOS ONE

Dear Dr. Dlamini, 

I'm pleased to inform you that your manuscript has been deemed suitable for publication in PLOS ONE. Congratulations! Your manuscript is now being handed over to our production team.

Kind regards, 

on behalf of

Dr. Raúl Alejandro Alegría-Morán 

Academic Editor

PLOS ONE